# A dual role for $Ca_v$1.4 $Ca^{2+}$ channels in the molecular and structural organization of the rod photoreceptor synapse

J Wesley Maddox[1,2,3†], Kate L Randall[1,2,3†], Ravi P Yadav[1], Brittany Williams[1,2,3‡], Jussara Hagen[1,2,3], Paul J Derr[4], Vasily Kerov[1,2], Luca Della Santina[5], Sheila A Baker[2,6,7], Nikolai Artemyev[1,2,7], Mrinalini Hoon[4,8], Amy Lee[1,2,3†]*

[1]Department of Molecular Physiology and Biophysics, University of Iowa, Iowa City, United States; [2]Iowa Neuroscience Institute, University of Iowa, Iowa City, United States; [3]Pappajohn Biomedical Institute, University of Iowa, Iowa City, United States; [4]Department of Neuroscience, University of Wisconsin-Madison, Madison, United States; [5]Department of Ophthalmology, University of California, San Francisco, San Francisco, United States; [6]Department of Biochemistry, University of Iowa, Iowa City, United States; [7]Department of Ophthalmology, Iowa City, United States; [8]Department of Ophthalmology and Visual Science, University of Wisconsin-Madison, Madison, United States

*For correspondence: amy.lee1@austin.utexas.edu

Present address: †Department of Neuroscience, University of Texas-Austin, Austin, United States; ‡Department of Cell Biology & Physiology, Carolina Institute for Developmental Disabilities, and Neuroscience Center, University of North Carolina, Chapel Hill, United States

Competing interests: The authors declare that no competing interests exist.

**Abstract** Synapses are fundamental information processing units that rely on voltage-gated $Ca^{2+}$ ($Ca_v$) channels to trigger $Ca^{2+}$-dependent neurotransmitter release. $Ca_v$ channels also play $Ca^{2+}$-independent roles in other biological contexts, but whether they do so in axon terminals is unknown. Here, we addressed this unknown with respect to the requirement for $Ca_v$1.4 L-type channels for the formation of rod photoreceptor synapses in the retina. Using a mouse strain expressing a non-conducting mutant form of $Ca_v$1.4, we report that the $Ca_v$1.4 protein, but not its $Ca^{2+}$ conductance, is required for the molecular assembly of rod synapses; however, $Ca_v$1.4 $Ca^{2+}$ signals are needed for the appropriate recruitment of postsynaptic partners. Our results support a model in which presynaptic $Ca_v$ channels serve both as organizers of synaptic building blocks and as sources of $Ca^{2+}$ ions in building the first synapse of the visual pathway and perhaps more broadly in the nervous system.

## Introduction

It is well established that the primary function of voltage-gated $Ca_v$ $Ca^{2+}$ channels within axon terminals is to serve as a conduit for $Ca^{2+}$ ions that trigger the fusion of neurotransmitter-laden vesicles with the presynaptic membrane (reviewed in *Dolphin and Lee, 2020*). Mutations in genes encoding presynaptic $Ca_v$ channels cause neurological disorders such as epilepsy, migraine, and ataxia (*Pietrobon, 2010*). When studied in animal models, these mutations lead to aberrant synaptic transmission and alterations in network excitability (*Tottene et al., 2009*; *Eikermann-Haerter et al., 2011*). Due to the steep $Ca^{2+}$-dependence of neurotransmitter release (*Dodge and Rahamimoff, 1967*), factors that modulate $Ca_v$ channel function can dramatically alter synaptic output. Thus, efforts to understand the pathophysiological consequences of disease-causing mutations have largely centered on how they affect biophysical parameters of channel opening, closing, or inactivation (*Pietrobon, 2010*).

However, $Ca_v$ channels are known to engage in pathways that do not involve their $Ca^{2+}$ conductance. For example, voltage-dependent changes in the conformation of $Ca_v$1.1 channels, rather than the incoming $Ca^{2+}$ ions, initiate skeletal muscle contraction (*Adams et al., 1990*). Despite its

importance for understanding fundamental cell biological processes in health and disease, the prevalence of non-conducting roles of $Ca_v$ channels in neurons is poorly understood. Analyses of $Ca_v$ knockout (KO) mice have helped illuminate the physiological functions of $Ca_v$ channels (*Pietrobon, 2005*), however, this approach does not distinguish between the $Ca^{2+}$-dependent and $Ca^{2+}$-independent contributions of these channels. Moreover, complex double and triple-KO approaches are needed given that most synapses in the nervous system utilize more than one $Ca_v$ subtype (*Dolphin and Lee, 2020*).

To overcome these hurdles, we focused on the $Ca_v$1.4 L-type channel, which is the predominant $Ca_v$ subtype in rod and cone photoreceptors in the retina (*Mansergh et al., 2005*; *Chang et al., 2006*; *Liu et al., 2013*). More than 140 mutations in the human *CACNA1F* gene encoding $Ca_v$1.4 have been identified, many of which cause heterogeneous forms of vision impairment (*Waldner et al., 2018*). The absence of photoreceptor synaptic responses in $Ca_v$1.4 KO mice has been attributed to a requirement for $Ca_v$1.4 in the presynaptic release of glutamate (*Mansergh et al., 2005*; *Chang et al., 2006*; *Liu et al., 2013*). However, an additional contributing factor is that photoreceptor synapses do not form in $Ca_v$1.4 KO mice (*Liu et al., 2013*; *Zabouri and Haverkamp, 2013*; *Regus-Leidig et al., 2014*). Does this developmental failure reflect a requirement for $Ca_v$1.4-mediated $Ca^{2+}$ signals or an alternative, non-conducting role for the $Ca_v$1.4 protein in organizing the photoreceptor synapse? Here, we generated a mouse strain that expresses a non-conducting mutant form of $Ca_v$1.4 to answer this question.

## Results and discussion

Rod and cone photoreceptors communicate visual information via a triadic 'ribbon' synapse containing postsynaptic neurites of a depolarizing (ON) bipolar cell and two horizontal cells (HCs) (*Masland, 2012*; *Figure 1A*). The ribbon, an organelle that critically regulates the properties of glutamate release, is a morphological hallmark of this and other sensory synapses (*Matthews and Fuchs, 2010*). During the first postnatal week in mice, ribbons develop from spherical precursors and are concentrated in the outer plexiform layer (OPL) where mature rod and cone synapses are localized (*Blanks et al., 1974*; *Regus-Leidig et al., 2009*). In $Ca_v$1.4 KO mice, the development of these synapses appears stunted in that spheres, rather than ribbons, are present at all developmental ages and often found in the outer nuclear layer (ONL) that normally contains only photoreceptor somas. The terminals of rods and cones (i.e., spherules and pedicles, respectively) (*Figure 1A*) of $Ca_v$1.4 KO mice are misshapen, and a number of proteins typical of mature synapses are absent or aberrantly localized (*Raven et al., 2008*; *Liu et al., 2013*; *Zabouri and Haverkamp, 2013*; *Regus-Leidig et al., 2014*). If $Ca_v$1.4 $Ca^{2+}$ signals are needed for the assembly of photoreceptor synapses, then ribbons and key synaptic proteins should be lacking from photoreceptors of a mouse strain expressing a non-conducting mutant form of $Ca_v$1.4, just as in $Ca_v$1.4 KO mice.

To test this prediction, we deployed a strategy based on a disease-causing mutation in the $Ca_v$1.3 L-type channel. The mutation results in the insertion of a glycine residue into transmembrane segment 6 of domain I (G369i, *Figure 1B*) and prevents $Ca^{2+}$ influx through the channel during membrane depolarizations (*Baig et al., 2011*). We determined if this mutation has a similar effect in $Ca_v$1.4 in electrophysiological recordings of transfected human embryonic kidney cells (HEK293T). The wild-type (WT) and G369i mutant channels were co-expressed with $\beta_2$ and $\alpha_2\delta-4$ subunits which are the major $Ca_v$ auxiliary subunits forming $Ca_v$1.4 complexes in photoreceptor synaptic terminals (*Lee et al., 2015*). G369i mutant channels were associated with the plasma membrane but did not mediate significant inward $Ba^{2+}$ currents ($I_{Ba}$) like WT channels (*Figure 1—figure supplement 1*). To functionally verify that G369i mutant channels reached the cell-surface, we analyzed ON gating currents ($I_{gating}$), which result from the movement of charged voltage-sensing domains prior to channel opening. For this purpose, we utilized the $\alpha_2\delta-1$ subunit which we find to produce larger current densities and better resolution of gating currents than with $\alpha_2\delta-4$. Compared to robust peak $I_{Ba}$ densities mediated by WT channels ($-20.70 \pm 5.31$ pA/pF), those in cells transfected with G369i mutant channels were negligible ($-1.14 \pm 0.11$ pA/pF, p=0.03) and not significantly different from that in cells transfected only with $\beta_2$ and $\alpha_2\delta-1$ as negative controls ($-0.56 \pm 0.15$ pA/pF, p=0.72, by Kruskal-Wallis test and post-hoc Dunn's test; *Figure 1C*). $I_{gating}$ was evident for both WT and G369i mutant channels, but not in the negative control cells. To estimate the number of channels with functional voltage-sensors in the membrane, we analyzed the time integral of $I_{gating}$ evoked at

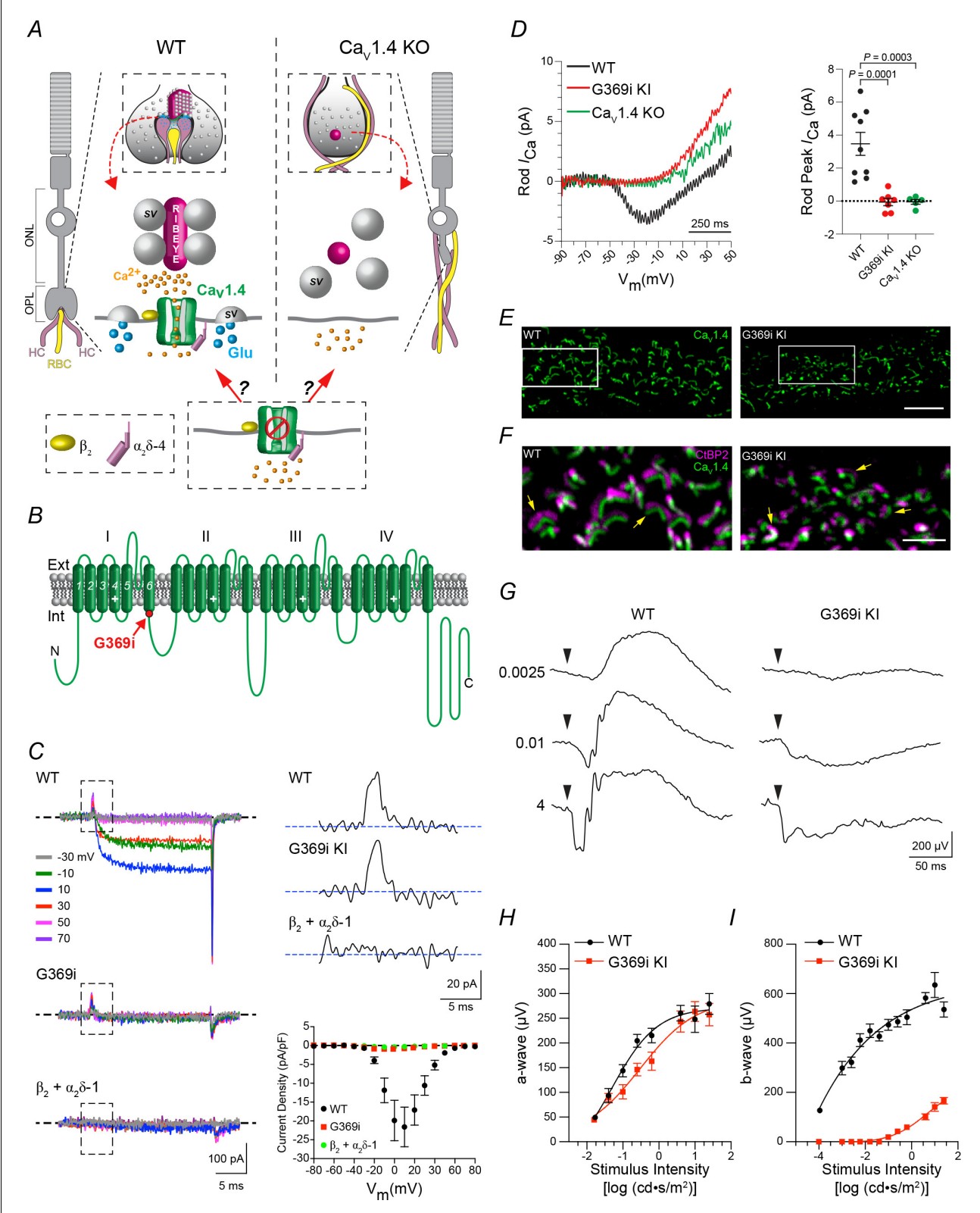

**Figure 1.** Characterization of G369i KI mouse strain to study the requirement of Ca$_v$1.4 Ca$^{2+}$ signals in rod synaptogenesis. (A), potential roles of Ca$_v$1.4 in the assembly of rod synapses. In the outer plexiform layer (OPL) of WT mice, neurites of two horizontal cells (HCs) and one rod bipolar cell (RBC) invaginate into the rod spherule; Ca$_v$1.4 clusters beneath the ribbon that tethers glutamate (Glu)-filled synaptic vesicles (SVs). In Ca$_v$1.4 KO retina, ribbons do not form, and HC and RBC neurites sprout into the outer nuclear layer (ONL). The efficacy of a non-conducting Ca$_v$1.4 mutant channel to

*Figure 1 continued on next page*

*Figure 1 continued*

support rod synaptogenesis will depend on whether $Ca^{2+}$ influx through $Ca_v1.4$ is required. (**B**), schematic of $Ca_v1.4$ pore-forming subunit with four homologous repeats (I–IV) each with six transmembrane domains; location of the glycine insertion in domain I (G369i) is indicated. G369i mutation (red). (**C**), *left*, representative $Ba^{2+}$ currents ($I_{Ba}$) evoked by 20 ms voltage steps from −90 mV to the indicated voltages in transfected HEK293T cells. Boxed regions indicate gating currents evident in cells co-transfected with $\beta_2 + \alpha_2\delta-1$ and WT or G369i mutant channels but not in cells transfected with $\beta_2$ and $\alpha_2\delta-1$ alone. *Upper right*, representative gating currents from boxed regions (*left*) elicited by a voltage step to +70 mV. *Lower right*, graph shows I–V relationship for cells co-transfected with $\beta_2 + \alpha_2\delta-1$ alone (n = 5), or co-transfected with WT (n = 9) or G369i mutant channels (n = 6). Points represent mean ± SEM. (**D**), representative traces (*left*) and average amplitudes of peak rod $Ca^{2+}$ currents ($I_{Ca}$, *right*) recorded in acute slices from WT, G369i KI, and $Ca_v1.4$ KO mouse retinas. Stimulus protocol was a 1 s voltage ramp from −90 to +50 mV. Points represent individual recorded rods with the mean and SEM indicated with bars. *P* values were determined by one-way ANOVA with Uncorrected Fisher's Least Significant Difference post-hoc test. (**E**), confocal micrographs of the OPL of wild-type (WT) and G369i KI retina double-labeled with antibodies against $Ca_v1.4$ and CtBP2 (CtBP2 labeling removed here for clarity). (**F**), expanded view of the boxed regions in *E* showing both CtBP2 and $Ca_v1.4$ labeling. Arrows depict arc-shaped ribbons apposed to $Ca_v1.4$ labeling. Scale bars, 5 μm. (**G**) representative voltage responses from flash ERGs recorded in dark- adapted WT (n = 4) and G369i KI (n = 4) mice. Arrowheads indicate time of flash. Numbers indicate flash intensities (cd•/$m^2$). (**H, I**), a-wave amplitudes (**H**) and b-wave amplitudes (**I**) measured from recordings obtained as in (**G**). Points represent mean ± SEM.

The online version of this article includes the following source data and figure supplement(s) for figure 1:

**Source data 1.** Traces corresponding to $I_{Ca}$ that were used for analyses of rod peak $I_{Ca}$ in *Figure 1D* are shown in the '.PDF' file.
**Figure supplement 1.** Characterization of G369i mutation in transfected HEK293T cells.
**Figure supplement 2.** Characterization of the G369i KI mouse line.

the reversal potential ($Q_{max}$, [*Wei et al., 1994*]). When normalized to cell size, $Q_{max}$ did not differ for WT (4.22 ± 1.34 fC/pF) and G369i mutant channels (3.22 ± 0.48 fC/pF, p=0.57 by unpaired t-test; *Figure 1C*). Thus, G369i prevents $Ca^{2+}$ conductance but not cell-surface trafficking or voltage-sensor movements in $Ca_v1.4$.

Armed with this knowledge, we incorporated the G369i mutation into $Ca_v1.4$ channels in vivo by genome editing in mice (*Figure 1—figure supplement 2*). In agreement with our findings in the heterologous expression system, inward currents such as those measured in rods of WT mice, were not evident in those of the knock-in mouse strain (G369i KI). The absence of depolarization-evoked currents was similar in G369i KI mice and a mouse strain completely lacking $Ca_v1.4$ expression ($Ca_v1.4$ KO, *Figure 1D*). However, $Ca_v1.4$ protein was detected in retinal lysates of G369i KI mice by Western blot (*Figure 1—figure supplement 2*). Moreover, immunofluorescence corresponding to G369i mutant channels was abundant in the OPL, where photoreceptor synapses normally form, and juxtaposed near the ribbon as in WT mouse retina (*Figure 1E*). Therefore, while it does not mediate voltage-dependent $Ca^{2+}$ currents, the G369i mutant channel is expressed and targeted appropriately in photoreceptors of G369i KI mice. Electroretinograms (ERGs) of dark-adapted G369i KI mice lacked b-waves at all but the highest light intensities, consistent with a disruption of rod synaptic transmission (*Figure 1I*). A-waves, which represent light-dependent hyperpolarization of photoreceptors, were not significantly different in WT and G369i KI mice (*Figure 1H*). Thus, G369i KI mice show defects in photoreceptor function that are largely restricted to synaptic terminals.

To define the contribution of $Ca_v1.4$ $Ca^{2+}$ signals to the organization of the photoreceptor synapse, we compared the molecular composition and structure of photoreceptor synapses in the retinas of WT, G369i KI, and $Ca_v1.4$ KO mice. We focused on rod terminals due to their simpler synaptic organization (i.e. one large, arc-shaped ribbon and a single type of postsynaptic bipolar cell) as compared to cones (i.e. multiple short ribbons and types of postsynaptic bipolar cells). Furthermore, rod synapses are more severely disrupted than cone synapses when the presynaptic abundance of $Ca_v1.4$ is compromised (*Ball et al., 2002*; *Katiyar et al., 2015*; *Wang et al., 2017*; *Kerov et al., 2018*). We first compared the prevalence of ribbons and spheres using antibody labeling for CtBP2, a component of the major ribbon protein Ribeye (*Schmitz et al., 2000*; *Figure 2A*). We restricted analyses to CtBP2-labeled structures in rod spherules by excluding those in cone arrestin-labeled cone pedicles. The minimum ribbon length was arbitrarily set at 0.95 μm based on the absence of values higher than this in $Ca_v1.4$ KO retina. While shorter on average than in WT retina, arc-shaped ribbons were plentiful in the OPL of G369i retina (*Figure 2B–D*) and co-clustered with proteins characteristic of mature rod synapses (*Figure 2E*) such as PSD-95 and β-dystroglycan (*Koulen et al., 1998*; *Blank et al., 1997*), which were undetectable in the OPL of $Ca_v1.4$ KO retina (*Figure 2F,G*). Bassoon, a protein involved in ribbon morphogenesis (*tom Dieck et al., 2005*), and RIM2, a protein

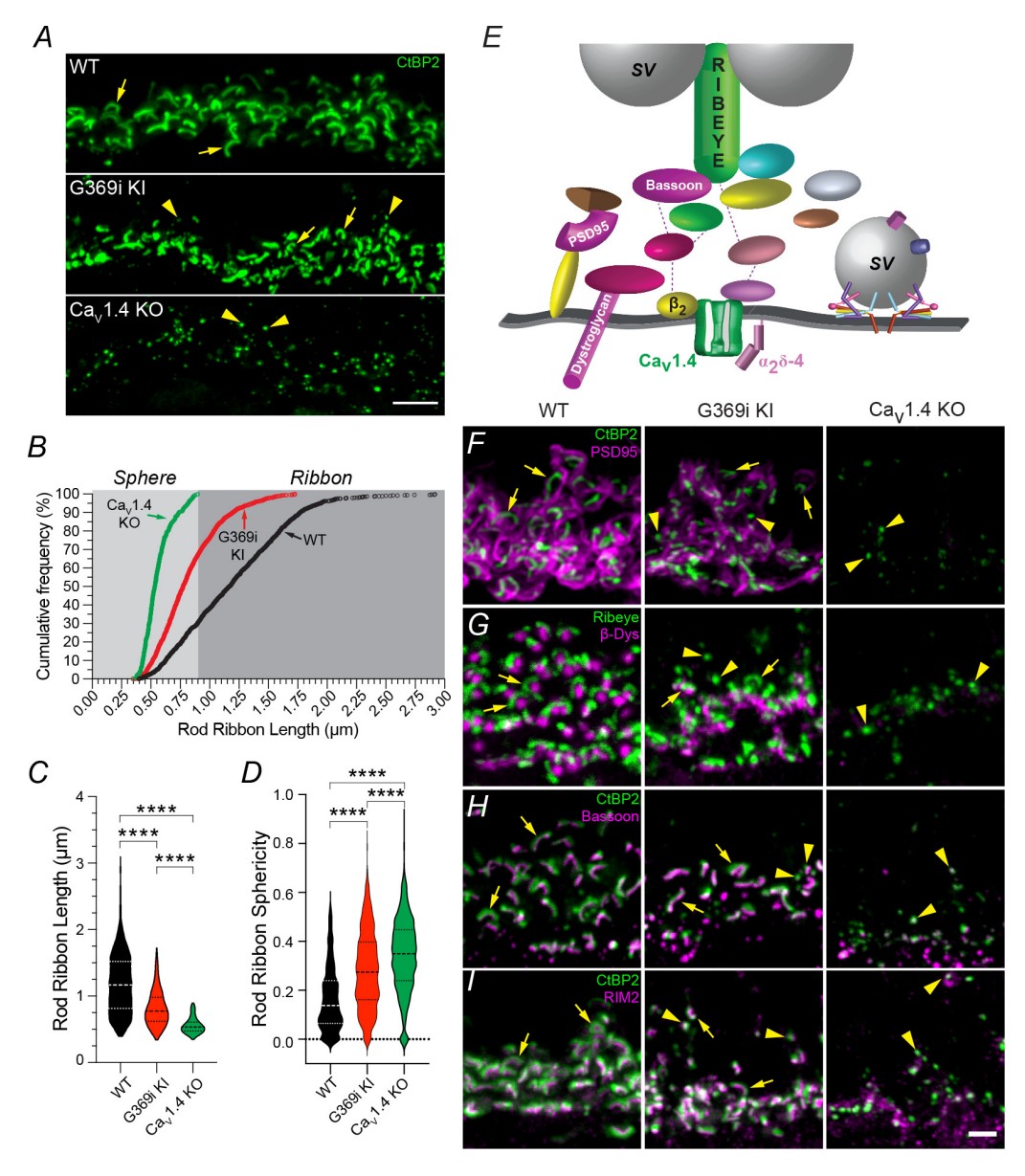

**Figure 2.** Rod ribbons still form in the absence of Ca$_v$1.4 Ca$^{2+}$ signals. (A) Confocal micrographs of the OPL of WT, G369i KI and Ca$_v$1.4 KO mice labeled with antibodies against CtBP2 to mark ribbons (arrows). Arrowheads depict spheres resembling immature ribbons. (B) Cumulative frequency of rod ribbon lengths in WT, G369i KI, and Ca$_v$1.4 KO. Ribbon states less than and greater than 0.95 μm were classified as spheres and ribbons, respectively. (C, D) Violin plots of rod ribbon lengths (C) and sphericity (D) obtained from dataset in B. **** p<0.001 by one-way ANOVA with Tukey's multiple comparison post-hoc test. WT, n = 1264 ribbons; KO, n = 348 ribbons; G369i KI, n = 1322 ribbons. Data were collected from the retinas of at least six individual mice per genotype. (E) Schematic of a rod active zone and associated proteins. SV, synaptic vesicle. (F–I) Confocal micrographs (maximum z-projections) of the OPL of WT, G369i KI and Ca$_v$1.4 KO mice double-labeled with antibodies against CtBP2 and PSD95 (F), Ribeye and β-dystroglycan (β-Dys), (G) CtBP2 and Bassoon (H) and CtBP2 and RIM2 (I). Arrows and arrowheads depict ribbons and spheres, respectively. Scale bars, 5 μm in A, 2 μm in F-I.

The online version of this article includes the following source data for figure 2:

**Source data 1.** The image archive contains all images used in quantitative analyses in *Figure 2B–D*.

implicated in regulating Ca$_v$1.4 at rod synapses (*Grabner et al., 2015*), were tightly clustered with ribbons in G369i KI as in WT retina (*Figure 2H,I*). Thus, although leading to shorter ribbons, the pre-synaptic assembly of rod synapses can proceed without Ca$_v$1.4 Ca$^{2+}$ signals.

We next determined if postsynaptic elements were also intact at G369i KI rod synapses. At the depolarized voltage of photoreceptors in darkness, Ca$_v$1.4-dependent glutamate release activates a metabotropic glutamate receptor (mGluR6), which suppresses the activity of TRPM1-dependent channels (*Shiells et al., 1981*; *Shen et al., 2009*). Light-dependent hyperpolarization of photorecep-tors reduces Ca$_v$1.4 Ca$^{2+}$ signals, which slows the rate of glutamate release, thus disinhibiting the postsynaptic conductance and enabling excitatory postsynaptic currents (EPSCs) that communicate light signals via the ON pathway (*Shiells et al., 1981*; *Figure 3A*). In agreement with previous work (*Koike et al., 2010*), mGluR6 was clustered postsynaptic to rod ribbons, and TRPM1 was situated at the tips of RBC dendrites in WT retina (*Figure 3B–D*). Although the subcellular distribution of the postsynaptic proteins was partially obscured by the diffuse pattern of TRPM1 labeling, arc-shaped ribbons in apposition to colocalized TRPM1 and mGluR6 could clearly be distinguished in G369i KI retina but not in Ca$_v$1.4 KO retina (*Figure 3B–D*). To test if this signaling pathway was functional in

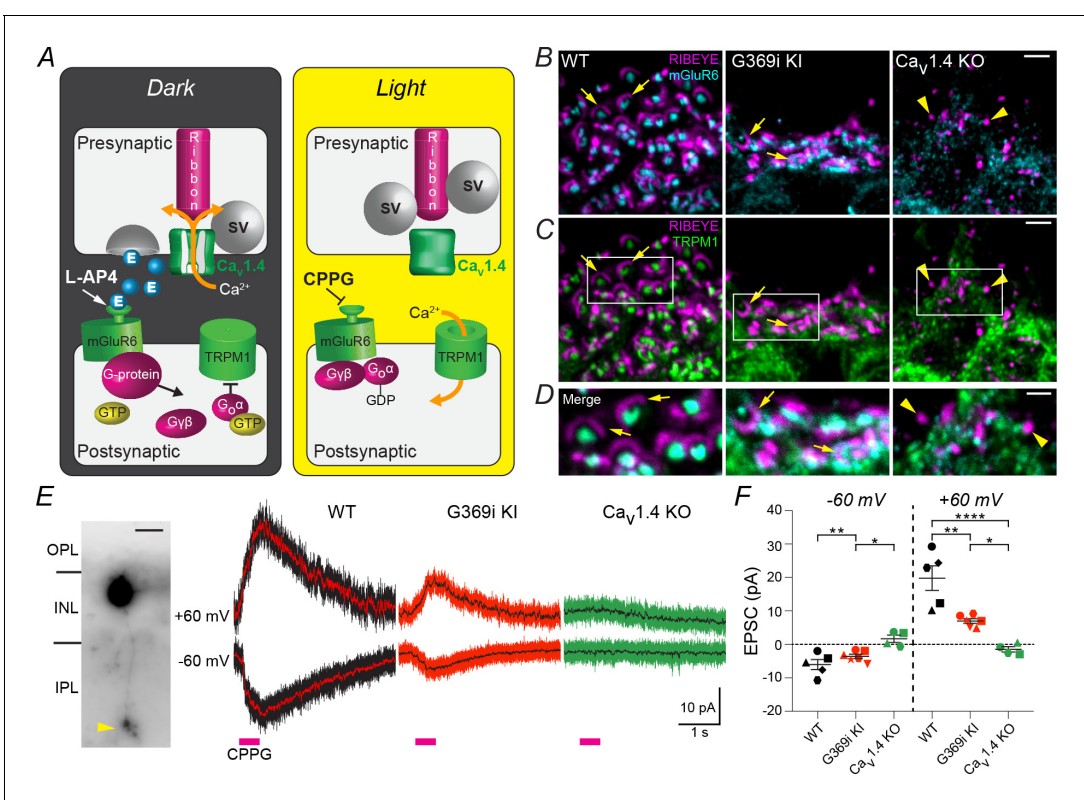

**Figure 3.** Postsynaptic signaling at rod synapses is partially intact in the absence of Ca$_v$1.4 Ca$^{2+}$ signals. (A), schematic illustrating synaptic transmission at rod-RBC synapses. In the dark, rods are depolarized, which activates Ca$_v$1.4 and Ca$^{2+}$-triggered glutamate (E) release. Activation of the mGluR6/G-protein pathway by glutamate or L-AP4 inhibits TRPM1-dependent channels. Light signals or the mGluR6 antagonist CPPG activate the ON pathway by disinhibiting the TRPM1-dependent conductance. (B–D), confocal micrographs of the OPL of WT, G369i KI and Ca$_v$1.4 KO mice triple-labeled with antibodies against Ribeye, mGluR6, and TRPM1. In B,C, only signals for Ribeye and mGluR6 (B) or TRPM1 (C) are shown for clarity. (C) Higher magnification views of boxed regions are shown in the merged image (D). Arrows and arrowheads depict ribbons and spheres, respectively. (E–F), whole-cell patch clamp electrophysiology of RBCs recorded in WT, G369i KI, and Ca$_v$1.4 KO mouse retinal slices. E, left, representative image of Lucifer Yellow-filled RBC showing typical lobular-shaped terminal (arrowhead). *Right*, representative current responses to CPPG puffs in RBCs held at −60 mV and +60 mV. Lines within each current trace are low-pass FFT filtered data. F, peak amplitudes of CPPG-evoked EPSCs. Each shape represents individual cells recorded at −60 and +60 mV. Bars represent mean ± SEM. ****p<0.0001, **p<0.01, *p<0.05, by one-way ANOVA with Fishers least significant difference post-hoc test. Scale bars, 2 µm in B,C, 1 µm in D, 15 µm in E.

The online version of this article includes the following source data for figure 3:

**Source data 1.** Traces corresponding to EPSCs that were used for analyses of EPSC amplitudes in *Figure 3F* are included in the '.PDF' file; individual values obtained for EPSC amplitudes in different cells are listed in the '.xlsx' file.

RBCs of G369i KI mice, we used an agonist and antagonist of mGluR6 (L-AP4 and CPPG, respectively) to simulate light responses in RBCs (*Shen et al., 2009*; *Figure 3A*). As expected (*Morgans et al., 2009*; *Shen et al., 2009*), the CPPG-evoked EPSC was robust and outwardly rectifying in WT RBCs (*Figure 3E,F*). Although significantly lower in amplitude compared to WT RBCs, this EPSC was measurable in all RBCs that were recorded in G369i KI mice but was never detected in any of the RBC recordings in $Ca_v1.4$ KO mice (*Figure 3E,F*). These results indicate that the molecular components of the postsynaptic signaling complex in RBCs can be assembled in the absence of $Ca_v1.4$ $Ca^{2+}$ signals, albeit less efficiently than in their presence.

Sprouting of HC and RBC neurites out of the OPL and into the ONL, and the appearance of ectopic ribbons and other presynaptic proteins in the ONL, occurs as a result of presynaptic functional impairments in rods and cones (*McCall and Gregg, 2008*). Thus, the prevailing view has been that synaptic transmission is needed to maintain the laminar organization of photoreceptor synapses in the OPL. Consistent with this notion, HC and RBC sprouting (*Figure 4A*) are particularly prominent in $Ca_v1.4$ KO mice, which lack evidence of photoreceptor transmission (*Chang et al., 2006*; *Raven et al., 2008*; *Liu et al., 2013*; *Zabouri and Haverkamp, 2013*). If these defects arise solely from the absence of $Ca_v1.4$ $Ca^{2+}$ signals that trigger glutamate release, the severity of neurite sprouting should be similar in G369i KI mice as in $Ca_v1.4$ KO mice. Contrary to this prediction, HC and RBC neurite sprouting was less prominent in G369i KI mice than in $Ca_v1.4$ KO mice *(Figure 4A, B)* with most of the synapses situated normally in the OPL (*Figure 4C*). These results indicate a supporting, but non-essential, role for $Ca_v1.4$ $Ca^{2+}$ signals and/or glutamatergic transmission for the proper lamination of rod synapses within the OPL.

Compared to the spherical appearance of ribeye-labeled structures in the OPL/ONL of $Ca_v1.4$ KO and other mouse strains with presynaptic dysfunction (*Chang et al., 2006*; *McCall and Gregg, 2008*; *Raven et al., 2008*; *Liu et al., 2013*; *Zabouri and Haverkamp, 2013*), those in the ONL of G369i KI mice exhibited mature ribbon morphology and were associated with synaptic proteins as in the WT OPL (*Figure 4C,D*). To further analyze the seemingly normal organization of these synapses in the OPL and ONL, we performed 3-dimensional (3D) reconstructions of spherules in G369i KI retina via serial block-face scanning electron microscopy (*Figure 4E–G*). Each of the four rod terminals sampled from G369i KI retina had a single ribbon that was anchored at the presynaptic membrane and was apposed to a triad of postsynaptic partners (*Figure 4F–G*). However, several peculiarities were observed. First, three of the four spherules that were analyzed contained more than one ribbon, rather than the single ribbon characteristic of rod terminals in the WT mouse retina (*Blanks et al., 1974*; *Figure 4E–G*). Second, some ribbons appeared club-shaped (*Figure 4—figure supplement 1*). Third, at least one of the extraneous ribbons was 'floating' rather than anchored near the presynaptic membrane. These alterations of ribbon architecture in G369i KI rod spherules resemble the types of ribbon plasticity caused by reductions in presynaptic $Ca^{2+}$ in mouse rods due to light or $Ca^{2+}$ chelators (*Spiwoks-Becker et al., 2004*; *Dembla et al., 2020*). Thus, the lack of $Ca_v1.4$ $Ca^{2+}$ signals in G369i KI rods could have caused fragmentation of the anchored ribbon, leading to the appearance of additional short, floating ribbons.

A particularly prominent anomaly of the reconstructed spherules was an alteration in postsynaptic architecture. Normally, within the invagination of mouse rod spherules are two HC neurites and an intervening RBC neurite (*Figure 4E*). Compared to HCs, the RBC neurite tip and the resident mGluR6 receptors are maintained relatively distant—hundreds of nanometers—from the release site (*Rao-Mirotznik et al., 1995*). This complex organization is thought to be critical for controlling glutamate release volume and spillover dynamics, allowing for differential activation of distinct glutamate receptor populations (*Petralia et al., 2018*). For spherules in the OPL as well as in the ONL of G369i KI retina, branching neurites of HCs and RBCs formed non-invaginating triadic contacts close to the anchored presynaptic ribbon (*Figure 4F,G*), rather than invading rod spherules as in WT retina (*Figure 4E*). Nevertheless, immunolabeling for mGluR6 remained clustered near ribbons in G369i KI retina (*Figure 3B*; *Figure 4—figure supplement 2*), suggesting that pre and postsynaptic apposition is maintained even in the absence of the normal spacing between RBC tips and the release sites. In agreement with these findings, mGluR6 labeling was significantly closer to that for ribbons in G369i KI than in WT retina as revealed by nearest-neighbor analyses (*Figure 4—figure supplement 2*). Thus, while dispensable for postsynaptic partner selection at rod synapses, $Ca_v1.4$ $Ca^{2+}$ signals are required for the invaginating arrangement of the corresponding neurites within the rod spherule and their proximity to release sites.

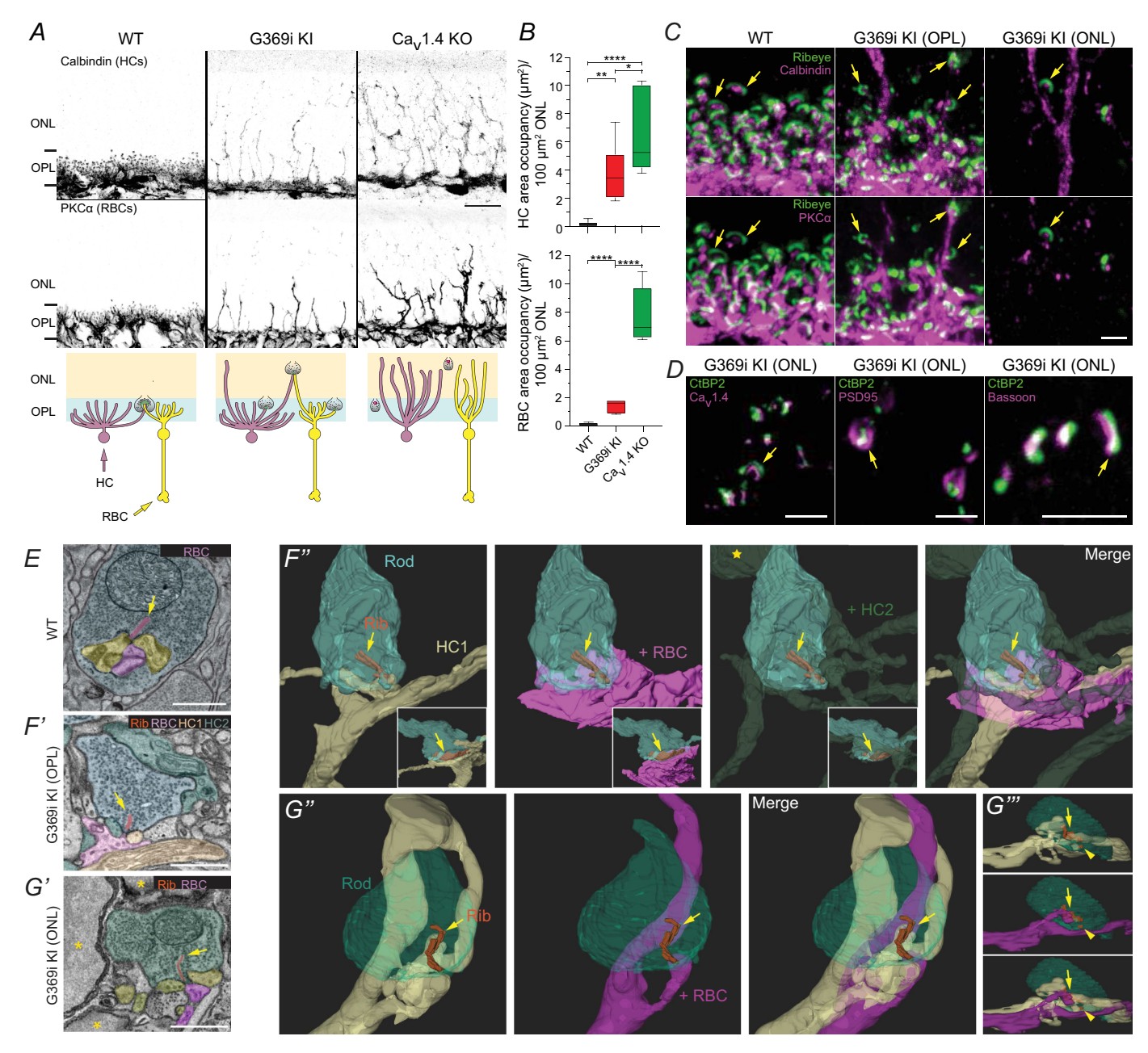

**Figure 4.** Rod synapses lack invaginating HC and RBC neurites in the absence of Ca$_v$1.4 Ca$^{2+}$ signals. (**A**), confocal micrographs of the ONL and OPL of WT, G369i KI and Ca$_v$1.4 KO retinas immunolabeled for calbindin and PKCα. The schematic below illustrates neurite sprouting in each genotype. (**B**), quantification of the area in the ONL occupied by HCs (calbindin) or RBCs (PKCα). One-way ANOVA with Fishers least significant difference post-hoc test. ****p<0.0001, **p<0.01, *p<0.05. n = 6 mice for each genotype. (**C**), confocal micrographs of the OPL of WT and the ONL and OPL of G369i KI retinas immunolabeled for Ribeye and Calbindin (*upper panels*) or PKCα (*lower panels*). (**D**), confocal micrographs of the ONL of G369i KI retinas that were immunolabeled for CtBP2 and Ca$_v$1.4 (*left*), PSD95 (*middle*) or Bassoon (*right*). (**E**), TEM image of a rod terminal in a WT retina. (**F,G**), serial block-face scanning electron microscopy images (**F',G'**) and 3D reconstructions (**F''–G'''**) of rod terminals in a G369i KI retina. Rod terminals located within the OPL (**F**) and ONL (**G**). The yellow star *F''* indicates a neurite from HC2 sprouting into the ONL. Insets in *F''* are a rotated view. *G'''* is a side view of *G''*. Arrows depict anchored ribbons. Arrowheads in G''' depict additional ribbon. Asterisks in G' indicate rod somas. Scale bars, 10 μm in *A*, 2 μm in *C-D*, 1 μm in *E,F',G'*. Rib, ribbon. RBC, rod bipolar cell. HC, horizontal cell.

The online version of this article includes the following source data and figure supplement(s) for figure 4:

**Source data 1.** The image archive contains all images used in quantitative analyses in *Figure 4B*.

**Figure supplement 1.** Multiple ribbons are found in rod terminals of G369i KI mice.

**Figure supplement 2.** Nearest-neighbor analysis of Ribeye- and mGluR6-labeled structures.

*Figure 4 continued on next page*

*Figure 4 continued*

**Figure supplement 2—source data 1.** The image archive contains all images used in quantitative analyses in *Figure 4—figure supplement 2B*; *Figure 4—figure supplement 2C*.

Like other presynaptic $Ca_v$ channels, $Ca_v$1.4 is expected to interact directly or indirectly with a diverse array of proteins (*Dolphin and Lee, 2020*), some of which are required for ribbon formation in photoreceptors (*tom Dieck et al., 2005*; *Kiyonaka et al., 2012*). Thus, the $Ca_v$1.4 complex could act as a central scaffold for protein interactions that drive the assembly of the ribbon and associated components of the active zone. Alternatively, but not mutually exclusively, voltage-dependent conformational changes in the $Ca_v$1.4 protein, which are intact in G369i mutant channels (*Figure 1C*), could be involved. For example, $Ca_v$1.4 could conformationally couple to intracellular $Ca^{2+}$ release channels, as $Ca_v$1.1 does in skeletal muscle (*Adams et al., 1990*), which could represent a $Ca^{2+}$ source that aids presynaptic development.

A limitation of our study is that the mutant G369i channel was expressed in all retinal cell-types that normally express $Ca_v$1.4. In addition to photoreceptors, $Ca_v$1.4 appears to be expressed in bipolar cells and at much lower levels in other cell-types in mouse retina (*Yan et al., 2020*). Because we find specific immunoreactivity for $Ca_v$1.4 in the outer retina to be limited to photoreceptor terminals (*Liu et al., 2013*), we favor the interpretation that $Ca_v$1.4 protein is not present to any large extent within bipolar cell soma or dendrites and that the postsynaptic structural defects of G369i KI rod synapses result primarily from the absence of presynaptic $Ca_v$1.4 $Ca^{2+}$ signals in rods. In addition to supporting glutamatergic transmission, presynaptic $Ca_v$1.4 $Ca^{2+}$ signals could strengthen trans-synaptic signaling via cell-adhesion molecules such as NGL-2 in HCs (*Soto et al., 2013*) and ELFN1 and pikachurin in RBCs (*Sato et al., 2008*; *Wang et al., 2017*), which collectively enable the invagination of the corresponding neurites into the rod terminal. Considering that rod ribbons are shorter in G369i KI than in WT retina (*Figure 2B*), $Ca_v$1.4 $Ca^{2+}$ signals may also regulate the activities of $Ca^{2+}$ sensitive proteins such as GCAP2, which binds to Ribeye and controls ribbon stability in a $Ca^{2+}$-dependent manner (*Schmitz, 2014*).

There are notable parallels and distinctions in the rod phenotypes of G369i KI mice and those lacking expression of either of the auxiliary subunits of $Ca_v$1.4, $\beta_2$ or $\alpha_2\delta-4$. $Ca_v\beta$ and $\alpha_2\delta$ subunits are known to facilitate the forward trafficking and maintenance of $Ca_v$ channels in the plasma membrane (*Dolphin and Lee, 2020*). Thus, it is not surprising that $\beta_2$ KO and $\alpha_2\delta-4$ KO mice exhibit severely diminished $Ca_v$1.4 $Ca^{2+}$ signals, and similar to G369i KI mice (*Figures 1G–I* and *4A–C*), there is evidence for a loss of rod synaptic transmission in ERGs, and significant sprouting of postsynaptic RBCs and HCs into the ONL in these mouse strains (*Ball et al., 2002*; *Katiyar et al., 2015*; *Wang et al., 2017*; *Kerov et al., 2018*). However, a striking difference is that ribbons and other aspects of the presynaptic complex are preserved in G369i KI but not in $\beta_2$ KO or $\alpha_2\delta-4$ KO mice (*Ball et al., 2002*; *Katiyar et al., 2015*; *Wang et al., 2017*; *Kerov et al., 2018*). Because we find significant levels of the mutant $Ca_v$1.4 protein within rod terminals of G369i KI mice (*Figure 1E, F*), which is not the case in $\beta_2$ KO or $\alpha_2\delta-4$ KO mice (*Ball et al., 2002*; *Katiyar et al., 2015*; *Wang et al., 2017*; *Kerov et al., 2018*), we propose that a minimal complement of $Ca_v$1.4 channels is needed to enable ribbon morphogenesis.

How synapses achieve their diverse forms and functions remains a fundamental question in neuroscience. Current evidence indicates that most synapses are assembled by intrinsic programs involving a variety of scaffolding proteins and cell-adhesion molecules (*Kurshan and Shen, 2019*), with ensuing patterns of neurotransmitter release needed primarily to instruct further refinements in synapse morphology and connectivity (*Sando et al., 2017*; *Sigler et al., 2017*). Our study identifies $Ca_v$1.4 as a nexus for both activity-dependent and independent mechanisms that drive the maturation of the rod synapse. Elucidating the signaling pathways that underlie the dual functions of $Ca_v$1.4 is an important challenge for future studies.

## Materials and methods

### Mice

All mouse protocols and procedures were approved by the University of Iowa Institutional Animal Care and Use Committee. To generate the G369i KI mouse, a CRISPR RNA (crRNA) was designed to target exon 7 of the mouse *Cacna1f* gene (5'-CTTGGAGTCCTAAGCGG). The repair template was designed to contain a glycine codon insertion corresponding to amino acid residue 369 with a silent point mutation to include the Fnu4HI restriction site (*Figure 1—figure supplement 2*). The Cas9 protein, synthetic crRNA, trans-activating crRNA (tracrRNA), and repair template (PAGE-purified donor oligo) were co-injected into pronuclei of mouse zygotes (C57BL6/J [RRID:IMSR_JAX:000664] × C57BL/6SJL[F1] [RRID:IMSR_JAX:100012]) by the University of Iowa Genome Editing Facility. Genotyping of mice was performed by PCR (forward primer 5'-TGCCTTGGGTGTACTTTGTG, reverse primer 5'-CCGAGCTCAGATGGAGTTTATG) followed by digestion of the PCR product with the Fnu4HI restriction enzyme. Mice were backcrossed to the C57BL/6J parental strain for 10 generations. Since the *Cacna1f* gene encoding Ca$_v$1.4 is on the X chromosome, only males of each genotype were used for experiments. WT littermates were used as controls. Ca$_v$1.4 KO (RRID:IMSR_JAX: 017761) mice were obtained from The Jackson Laboratory.

### Immunohistochemistry

Mice at postnatal day 42 (P42) were euthanized with a lethal dose of isoflurane followed by cervical dislocation. Eyes were enucleated and hemisected. The remaining eye cups were fixed in ice-cold 4% paraformaldehyde for 30 min. Fixed eye cups were washed three times with 0.1 M PB containing 1% glycine. Eye cups were infused with 30% sucrose at 4°C overnight and frozen in a 1:1 (wt/vol) mixture of Optimal Cutting Temperature compound (OCT, Sakura Finetek, Torrence, CA) and 30% sucrose in a dry ice/isopentane bath. Eye cups were cryosectioned at 20 μm on a Leica CM1850 cryostat (Leica Microsystems, Wetzlar, Germany), mounted on Superfrost plus Micro Slides (VWR, Radnor, PA), dried for 5 to 10 min at 42°C, and stored at −20°C. Slides with mounted cryosections were warmed to room temperature, washed with 0.1 M PB for 30 min to remove the OCT/sucrose mixture, and blocked with dilution solution (DS, 0.1 M PB/10% goat serum/0.5% Triton-X100) for 15 min at room temperature. All remaining steps were carried out at room temperature. All primary antibodies and appropriate secondary antibodies were diluted in DS at concentrations specified in *Table 1*. Sections were incubated with primary antibodies for 1 hr or overnight and then washed five times with 0.1 M PB. Sections were then incubated with secondary antibodies for 30 min and then washed five times with 0.1 M PB. Labeled sections were mounted with #1.5 coverslips (Leica) using Fluoromount-G (Electron Microscopy Sciences, Hatfield, PA).

**Table 1.** Antibodies used in this study.

| Antibody | Host/clonality | Manufacturer | Cat no. | RRID |
|---|---|---|---|---|
| Bassoon | ms-IgG2A | Thermo Fisher Scientific | MA1-20689 | AB_2066981 |
| Cone Arrestin | rabbit | Millipore | AB15282 | AB_1163387 |
| Calbindin (D-28K) | mouse-IgG1 | Sigma | C9848 | AB_476894 |
| PKCα | mouse-IgG2A | Invitrogen | MA1-157 | AB_2536865 |
|  | rabbit | UC Santa Cruz | SC-208 | AB_2168668 |
| Ctbp2 | mouse-IgG1 | BD Biosciences | 612044 | AB_399431 |
| Ribeye | rabbit | Synaptic Systems | 192 103 | AB_2086775 |
| Psd95 | mouse-IgG2A | UC Davis/NIH NeuroMab | 75–028 | AB_2292909 |
| RIM2 | rabbit | Synaptic Systems | 140–103 | AB_887776 |
| mGluR6-366 | mouse | Dr. Theodore Wensel | N/A | N/A |
| TRPM1 | mouse-IgG1 | Dr. Theodore Wensel | N/A | N/A |
| Ca$_v$1.4 | rabbit | Lee Lab Ab167 | N/A | AB_2650487 |

## Image acquisition

Retinas that had been immunofluorescently labeled were visualized using an upright Olympus FV1000 confocal microscope (Tokyo, Japan) equipped with an UPLFLN 100x oil immersion objective (1.3 NA). Images were captured using the Olympus FluoView software package. Acquisition settings were optimized using a saturation mask to prevent signal saturation prior to collecting 16-bit, confocal z-stacks with voxel size of $0.09~\mu m \times 0.09~\mu m \times 0.2~\mu m (X \times Y \times Z)$. All confocal images presented are maximum z-projections.

## Molecular biology and transfection

The cDNAs used were $Ca_v1.4$ (GenBank: A930034B14), $\beta_{2a}$ (GenBank: NM_053851), $\beta_{2X13}$ (GenBank: KJ789960), and $\alpha_2\delta-4$ (GenBank: NM_172364) in pcDNA3.1 (*Lee et al., 2015*). To create the $Ca_v1.4$ cDNA with the G369i mutation, a codon insertion corresponding to the glycine insertion at residue G369 was generated with the HiFi DNA Assembly Cloning System (New England Biolabs) using $Ca_v1.4$ cDNA as the PCR template and the primers a1FbegNotF: 5'-TATATGCGGCCGCCCACCA TGGATTAC-3', Fr1G369insR: 5'-CTTAGGACACCTCCAAGCACAAGGTTGAGG-3', Fr2G369insF: 5'-TTGGAGGTGTCCTAAGCGGGGAGTTC-3', a1FBsrGIr: 5'-TTTAGGCAGCGTGTACAGCTAGCCA TGGTCC-3'. All constructs were verified by DNA sequencing before use. Human embryonic kidney 293 T (HEK293T) cells (American Type Culture Collection, Manassas, Virginia) were cultured in Dulbecco's Modified Eagle's Medium (Thermo Fisher Scientific, Waltham, MA) with 10% FBS (VWR) at 37°C in 5% $CO_2$. At 70–80% confluence, the cells were co-transfected with cDNAs encoding mouse $Ca_v1.4$ (1.8 μg) $\beta_{2X13}$ (0.6 μg), $\alpha_2\delta-4$ (0.6 μg), and enhanced GFP in pEGFP-C1 (0.1 μg) using FuGENE six transfection reagent (Promega, Madison, WI) according to the manufacturer's protocol. Cells treated with the transfection mixture were incubated at 37°C for 24 hr. Cells were then incubated at 30°C for an additional 24 hr before beginning experiments.

## Solutions

All reagents were purchased from Sigma-Aldrich unless otherwise stated. HEK293T external recording solution contained the following (in mM): 140 Tris, 20 $BaCl_2$, 1 $MgCl_2$, pH 7.3. HEK293T internal recording solution contained the following: 140 NMDG, 10 HEPES, 2 $MgCl_2$, 2 Mg-ATP, 5 EGTA, pH 7.3. The retina slicing solution was continuously bubbled with 100% $O_2$ and contained the following: Ames' Medium with L-glutamine supplemented with (in mM): 15 NaCl, 10 HEPES, 80 U/mL penicillin, 80 μg/mL streptomycin, pH 7.4. The extracellular slice recording solution was continuously bubbled with 95% $O_2$/5% $CO_2$ and contained the following: Ames' Medium with L-glutamine, 5 NaCl, 23 $NaHCO_3$, 80 U/mL penicillin, 80 μg/mL streptomycin. The intracellular pipette solution for recording rod photoreceptor $Ca^{2+}$ currents ($I_{Ca}$) contained the following: 120 $CsMeSO_4$, 10 tetraethylammonium (TEA) Cl, 1 $CaCl_2$, 1 $MgCl_2$, 5 Mg-ATP, 0.5 Na-ATP, 5 phosphocreatine, 10 HEPES, 2 ethylene glycol-bis(β-aminoethyl ether)-N,N,N',N'-tetraacetic acid (EGTA), 2 N-(2,6-Dimethylphenylcarbamoyl-methyl)triethylammonium bromide (QX 314 bromide), 0.1 mg/mL Lucifer Yellow, pH 7.35. The intracellular pipette solution for recording rod bipolar cells contained the following: 105 $CsMeSO_4$, 20 TEA-Cl, 4 Mg-ATP, 0.4 Na-ATP, 10 phosphocreatine, 10 HEPES, 2 1,2-bis(o-aminophenoxy)ethane-N,N,N',N'-tetraacetic acid (BAPTA), 10 glutamic acid, 0.1 mg/mL Lucifer Yellow, pH 7.35.

The mGluR6 agonist L-(+)−2-Amino-4-phosphonobutyric acid (L-AP4, Tocris) and antagonist (*RS*)-α-Cyclopropyl-4-phosphonophenylglycine (CPPG, Tocris) were dissolved in 4 mM NaOH and 100 mM NaOH, respectively. L-AP4 and CPPG were diluted into extracellular slice recording solution and retina slicing solution, respectively. The pH of each solution was checked after the addition of L-AP4 or CPPG and adjusted if necessary.

## Patch clamp electrophysiology

Transfected HEK293T whole-cell voltage clamp recordings were performed 48 to 72 hr after transfection using an EPC-9 amplifier and Patchmaster software (HEKA Elektronik, Lambrecht, Germany). Patch pipette electrodes with a tip resistance between 2 and 4 MΩ were pulled from thin-walled borosilicate micropipettes (VWR) using a P-97 Flaming/Brown Puller (Sutter Instruments, Novato, CA). A reference Ag/AgCl wire was placed into the culture dish mounted on an inverted Olympus IX70 microscope. Recordings were performed at room temperature (22–24°C). The series resistance was compensated up to 70%, and passive membrane leak subtraction was conducted using a p/−4

protocol. Whole-cell $I_{Ba}$ of transfected HEK293T cells were evoked for 50 ms with incremental 10 mV steps from −80 mV to +80 mV from a holding voltage of −90 mV. Data were analyzed using IgorPro 6.0 (WaveMetrics, Portland OR, USA).

Whole-cell voltage clamp recordings of rods and RBCs were performed using and EPC-10 amplifier and Patchmaster software (HEKA). Patch pipette electrodes with a tip resistance between 12 and 15 MΩ for rods and between 7 and 9 MΩ for RBCs were pulled from thick-walled borosilicate glass (1.5 mm outer diameter; 0.84 mm inner diameter; World Precision Instruments, Sarasota, FL, USA) using a P-97 Flaming/Brown Puller (Sutter Instruments, Novato, CA). To prepare retinal slices, adult mice (6–8 weeks old) were euthanized using a lethal dose of isoflurane followed by cervical dislocation. Eyes were enucleated, placed into cold Ames' media slicing solution, and hemisected. Following removal of the vitreous, the retina was separated from the back of the eye. Central retina was isolated, molded into low-melt agarose (Research Products International, Mount Prospect, IL, USA) and mounted in a Leica VT1200s vibratome (Leica Biosystems). Vertical retinal slices (~250 µM) were anchored in a recording chamber, placed onto a fixed stage (Sutter Instruments), and positioned under an upright Olympus BX51WI microscope with a water-immersion objective 40X (0.8 NA). A custom gravity perfusion system was used to deliver extracellular slice recording solution at a flow rate of 2 ml/min. Cell bodies of rods and RBCs identified based on their morphology and location using IR-DIC optics and a CCD Camera (Hamamatsu Photonics, Shizuoka, Japan) controlled with µManager software (doi:10.14440/jbm.2014.36). Cells were filled with Lucifer Yellow during each recording, and the morphology of each cell recorded was confirmed post hoc using wide-field fluorescence. A reference Ag/AgCl pellet electrode was placed in 3 M KCl and connected to the recording chamber with a 3 M KCl agarose bridge. Recordings were performed at room temperature (22–24℃). Data from whole-cell recordings with a series resistance >20 MΩ were discarded. Passive membrane leak currents present in voltage-ramp recordings were linearly subtracted post-hoc from rod $I_{Ca}$ using OriginLab (Northampton, Massachusetts, USA).

## Electroretinography

ERG recordings were obtained using the Espion E[3] system (Diagnosys LLC, Lowell, MA) as described previously (*Liu et al., 2013*). Following overnight dark adaptation, males (5–8 week-old) were anesthetized with a ketamine/xylazine mixture (100 mg/kg, i.p.). The pupils were dilated by applying a drop of 1% tropicamide, followed by a drop of 2.5% phenylephrine hydrochloride. ERGs were recorded simultaneously from the corneal surface of each eye using gold ring electrodes (Diagnosys), with an electrode placed on the back of the head as reference and another electrode placed near the tail as ground. A drop of Hypromellose 2.5% Ophthalmic Demulcent Solution (# 17478-0064-12, Akorn Gonak) was placed on the corneal surface to ensure electrical contact and to prevent eyes from drying and cataract formation. Body temperature of mice was maintained at 37℃ using the system's heating pad. Mice were placed in a Ganzfeld stimulator chamber (ColorDome; Diagnosys) for delivery of stimuli, and the mouse head and electrode positioning were monitored on the camera attached to the system. ERG responses were evoked by a series of flashes ranging from 0.0001 to 100 cd•s/m$^2$. Responses to six sweeps were averaged for dim flashes up to 0.6 cd•s/m$^2$, two sweeps were averaged for 4 cd•s/m$^2$, and responses to brighter flashes were recorded without averaging. Intersweep intervals for flashes with increasing strength were increased from 10 s to 60 s to allow full recovery from preceding flashes. To record photopic ERGs, mice were exposed to a background light (30 cd•s /m$^2$) for 3 min before flash stimulation (3, 30 or 100 cd•s/m$^2$); six sweeps were averaged for further analysis.

## Serial block-face scanning electron microscopy and 3D reconstructions

Eye cups were prepared from P42 WT and G369i KI littermates and fixed using 4% glutaraldehyde in 0.1M cacodylate buffer, pH 7.4, for 4 hr at room temperature followed by additional fixation overnight at 4℃. Glutaraldehyde-fixed eye cups were then washed 3 times in 0.1M cacodylate buffer. Retinas were thereafter isolated and embedded in Durcupan resin after staining, dehydration, and embedding as described previously (*Della Santina et al., 2016*). A Thermo Scientific VolumeScope serial block-face scanning electron microscope was used to image embedded retinas. Retinal regions comprising a 2 × 2 montage of 40.96 µm tiles were imaged at a resolution of 5 nm/pixel and section thickness of 50 nm. Image stacks were aligned, and rod photoreceptor terminals reconstructed using

TrakEM2 (NIH). Postsynaptic partners at the rod triad were followed to the INL to determine their identity (HC vs bipolar cell). Amira (Thermo Fisher Scientific) software was used for 3D visualization of reconstructed profiles.

### Quantification, statistical analysis, and experimental design

Immunofluorescent objects were detected within in z-stacks of 30 optical sections and quantified with ObjectFinder software (https://lucadellasantina.github.io/ObjectFinder/) using unbiased iterative threshold search algorithms. To measure rod ribbon length, sections were double-labeled with antibodies against CtBP2 and cone arrestin as described above. 3D binary masks of cone arrestin labeling were generated in order to exclude CtBP2 labeling within cones from analyses. The length of each rod ribbon was measured along the major axis. The sphericity of each detected ribbon was also calculated. Ribbons detected along the edges of the cone arrestin mask and the outer edges of the 3D images were excluded from analyses. To measure nearest-neighbor distances, ObjectFinder was used to detect and measure the distances from ribeye-labeled 'home' objects to the nearest mGluR6-labeled 'neighbor' objects.

To measure neurite occupancy of the outer nuclear area (ONL) by HCs, rod bipolar cells (RBCs), and cone bipolar cells (CBCs), sections were labeled with antibodies against calbindin (HCs), PKCα (RBCs), and secretagogin (CBCs). Confocal z-stacks were collapsed and labeling was converted to binary masks. Particle analysis (ImageJ) of labeling within the ONL area 12 µm above the OPL/INL border was performed, and occupancy was calculated as a function of the total area analyzed. Images used for analyses were collected from central retina from at least six animals/genotype.

Sample sizes were estimated based on those used previously to identify statistically significant differences in similar datasets and utilizing similar biological replicates (i.e. animals, transfections). All data presented result from at least three independent experiments. Data presented on graphs displayed as symbols and bars represent mean ± SEM. Statistical analyses were performed using GraphPad Prism (v 8.0) and a $p$ value of < 0.05 was considered significant. Outliers were identified using ROUT method (Q = 1%) using GraphPad Prism and were excluded from analyses. The specific $n$ for each experiment as well as the post-hoc test and corrected $p$ values, can be found in the figure legends.

## Acknowledgements

The authors thank Sharm Knecht and Rachel Wong (U.Washington) for assistance with serial block-face scanning electron microscopy image collection and comments on the manuscript; and Ted Wensel for providing mGluR6 and TRPM1 antibodies.

## Additional information

### Funding

| Funder | Grant reference number | Author |
| --- | --- | --- |
| National Eye Institute | EY 026817 | Amy Lee |
| McPherson Eye Research Institute | | Mrinalini Hoon |
| Research to Prevent Blindness | | Mrinalini Hoon |
| National Eye Institute | EY 029953 | J Wesley Maddox |
| National Eye Institute | EY 026477 | Brittany Williams |
| National Eye Institute | EY010843 | Nikolai Artemyev |
| National Eye Institute | EY012682 | Nikolai Artemyev |

The funders had no role in study design, data collection and interpretation, or the decision to submit the work for publication.

## Author contributions
J Wesley Maddox, Conceptualization, Formal analysis, Funding acquisition, Investigation, Writing - original draft, Writing - review and editing; Kate L Randall, Ravi P Yadav, Nikolai Artemyev, Formal analysis, Investigation, Writing - review and editing; Brittany Williams, Formal analysis, Funding acquisition, Investigation, Writing - review and editing; Jussara Hagen, Investigation, Writing - review and editing; Paul J Derr, Vasily Kerov, Investigation, Gave final approval of the version; Luca Della Santina, Software, Writing - review and editing; Sheila A Baker, Resources, Writing - review and editing; Mrinalini Hoon, Formal analysis, Funding acquisition, Visualization, Writing - review and editing; Amy Lee, Conceptualization, Resources, Data curation, Formal analysis, Supervision, Funding acquisition, Investigation, Writing - original draft, Project administration, Writing - review and editing

## Author ORCIDs
J Wesley Maddox (iD) https://orcid.org/0000-0002-1630-2746
Kate L Randall (iD) https://orcid.org/0000-0002-7199-9806
Amy Lee (iD) https://orcid.org/0000-0001-8021-0443

## Ethics
Animal experimentation: This study was performed in strict accordance with the recommendations in the Guide for the Care and Use of Laboratory Animals of the National Institutes of Health. All of the animals were handled according to approved institutional animal care and use committee (IACUC) protocols (#7121262-025) of the University of Iowa. The protocol was approved by the Office of Institutional Animal Care and Use Committee of the University of Iowa (A3021-01).

## Decision letter and Author response
Decision letter https://doi.org/10.7554/eLife.62184.sa1
Author response https://doi.org/10.7554/eLife.62184.sa2

# Additional files
## Supplementary files
• Transparent reporting form

## Data availability
All data generated or analysed during this study are included in the manuscript and supporting files.

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
