## [Decision Letter]

**Acceptance summary:**

This paper provides convincing evidence that presynaptic calcium channel proteins play an organizing role during synapse development, additionally to calcium influx and activity. This work provides insights into the cellular elements involved in synaptic structure and organization of rod ribbon synapses, with relevant implications in the study of development at other synapses.

**Decision letter after peer review:**

[Editors’ note: the authors submitted for reconsideration following the decision after peer review. What follows is the decision letter after the first round of review.]

Thank you for submitting your work entitled "A dual role for Ca_v_1.4 Ca^2+^ channels in the molecular and structural organization of the rod photoreceptor synapse" for consideration by *eLife*. Your article has been reviewed by three peer reviewers, and the evaluation has been overseen by a Reviewing Editor and a Senior Editor. The following individuals involved in review of your submission have agreed to reveal their identity: Joshua H Singer (Reviewer #2); Wallace B Thoreson (Reviewer #3). Our decision has been reached after consultation between the reviewers. Based on these discussions and the individual reviews below, we regret to inform you that your work cannot be considered for publication in *eLife* in its current form.

The main reason for our decision is related to issues that critically affect the main conclusions of the manuscript. Reviewer #2 raised a serious concern about the lack of an essential control experiment needed to demonstrate that the KI rods do generate light responses and that there is a deficit in transmission at the rod synapse. This issue was discussed during the consultation process, and reviewer #3, the Senior Editor and myself agreed with reviewer #2 that this experiment is indispensable in order for this work to be considered for publication in *eLife*. However, we would be willing to consider a new manuscript including this experiment and addressing all the other concerns expressed by the reviewers. Please keep in mind that this would be considered a new submission.

Reviewer #1:

The Ca_v_1.4 calcium channel is the major calcium channel in the OPL of the retina, which drive neurotransmitter release from both rods and cones. Ca_V_1.4 knockout mice have presynaptic and postsynaptic morphological abnormalities in the retina that had previously been described. This manuscript reports the use of a knock in mouse model to disambiguate the effects of the Ca_v_1.4 Ca^2+^ channel from those of calcium itself on synapse structure in the OPL. To do so, the authors make use of a KI mouse where the WT Ca_v_1.4 channel is replaced by a channel with a point mutation that renders it unable to conduct calcium. Interestingly, the authors find a small change in the size of and relatively moderate disorganization of presynaptic structures in the KI mouse model, more significant disruptions were observed in postsynaptic cells. The authors conclude that the Ca^2+^ channel protein is needed for proper formation and maintenance of the pre-synaptic structure, but calcium entry through channels are necessary for proper recruitment of post-synaptic partners. Overall, the results are novel and the conclusion is well supported by the data.

Specific comments:

1) The experiments in this paper focused on the synapses in rods. In WT, rod ribbons are easily distinguishable from cone ribbons based on size and shape. In the KI model, rod ribbons become smaller and one cannot assume that they will maintain their normal shape. Indeed, the authors describe multiple ribbons (some misshapen) in some rods in EM. Is the assumption that all spots are rod ribbons in the KO and KI, but only the crescent shaped ones are rod ribbons in the WT? If some are mis-assigned, how might this affect the quantification of the effects on ribbon size. If they can't be convincingly distinguished, then wouldn't it be fairer comparison to analyze all ribbons in the OPL regardless of genotype?

2) If there is a reliable way to distinguish rod ribbons from cone ribbons, there seems to be a missed opportunity here to look at effects on cone ribbons and their post-synaptic partners, at least at the level of IHC.

Reviewer #2:

This manuscript by Maddox et al. presents a study of Ca_v_1.4 in retinal synapse development. The central issue addressed is differentiating the signaling / organizational role(s) of the channel protein from those of Ca^2+^ influx through the channel during retinal rod photoreceptor synapse development. The authors generated a non-conducting channel (G369i KI) that appears to be trafficked normally to synapses and assessed differences between G369i mutants, Ca_v_1.4 KO, and wild-type animals. They found substantial differences between G369i mutants and Ca_v_1.4 KO retinas, with the G369i mutant more closely resembling the wild-type. The authors suggest, then, that the channel protein serves an organizing role during synapse development.

One critical control experiment has not been performed, and this must be included in any revision of the manuscript: although the authors show that rods themselves lack Ca^2+^ currents (Figure 1D), the authors do not provide evidence that the G369i KI rods respond to light and that postsynaptic rod bipolar cells do not (indicating a true deficit of synaptic transmission at the rod to rod bipolar cell synapse). As illustrated in Figure 3, the postsynaptic rod bipolar cells do appear to have developed mGluR6-mediated signaling cascades, but the authors' interpretation of the result, that the postsynaptic signaling can develop in the complete absence of presynaptic glutamate release, is not supported by the data presented. As well, the efficiency of Ca^2+^-exocytosis coupling at the rod presynapse is incredibly high, and it is worth noting that Ca^2+^ from sources other than voltage-gated Ca^2+^ channels could give rise to enough glutamate release to modulate synapse development.

With regard to the pre- and postsynaptic organization of the synapse, as illustrated in Figure 4: it seems a bit counterintuitive to me that the absence of an invaginating synapse would result in a closer apposition of ribeye and mGluR6 proteins, given that mGluR6 normally is located at the dendritic tips inserted in the very center of the invagination. The authors should clarify their explanation of this observation. As well, it is worth commenting that the authors addressed the integrity of the presynaptic active zone by visualizing ribeye and bassoon proteins using fluorescence immunohistochemistry, but neither protein-to my knowledge-interacts directly with Cavs. Is there a reason that the authors did not examine Rim / RimBP or CAST/ELKS expression as an assay of active zone organization? These proteins interact more directly with the Ca^2+^ channels.

Presumably, the G369i KI mutation should not affect the voltage-dependent gating of the channel. Has gating in the mutant been characterized? Is the voltage-sensing role of the channel important in development? These are interesting-to me, at least-questions that might be addressed experimentally or, at a minimum, in a revised discussion.

Reviewer #3:

This well-written, carefully done study shows that structural features, not calcium influx, of Ca_V_1.4 L-type calcium channels are sufficient for proper organization of many key molecules at rod ribbon synapses. However, formation of the synaptic invagination and proper arrangement of post-synaptic partners requires calcium influx. To perform this study, the authors created knockin mice that express a non-conducting form of Ca_V_1.4. The study was technically proficient and clearly described, with helpful illustrations accompanying each figure. Further work will be needed to understand which particular structural elements are important and how calcium influx regulates formation of the invaginating synapse. Testing the ability of rods to respond to light and transmit those signals to rod bipolar cells is important for interpreting their results. One relatively straightforward way to test this would be to record the A- and B-waves of the electroretinogram (ERG) in their mice.

---

## [Author Response]

[Editors’ note: the authors resubmitted a revised version of the paper for consideration. What follows is the authors’ response to the first round of review.]

The main reason for our decision is related to issues that critically affect the main conclusions of the manuscript. Reviewer #2 raised a serious concern about the lack of an essential control experiment needed to demonstrate that the KI rods do generate light responses and that there is a deficit in transmission at the rod synapse. This issue was discussed during the consultation process, and reviewer #3, the Senior Editor and myself agreed with reviewer #2 that this experiment is indispensable in order for this work to be considered for publication in eLife. However, we would be willing to consider a new manuscript including this experiment and addressing all the other concerns expressed by the reviewers. Please keep in mind that this would be considered a new submission.

We thank the reviewers for their careful assessment of our manuscript. We have addressed the primary concern that we perform a critical experiment to demonstrate that the KI rods do generate light responses and that there is a deficit in transmission at the rod synapse. We have addressed this concern as suggested by Reviewer 3 via electroretinograms (ERGs). These experiments show that while KI rods exhibit normal responses to light (*i.e*., normal a-waves), they show no evidence of b-waves that reflect transmission from rods to rod bipolar cells (RBCs). We have addressed these and other concerns raised by the reviewers as described below.

Reviewer #1:The Ca_v_1.4 calcium channel is the major calcium channel in the OPL of the retina, which drive neurotransmitter release from both rods and cones. Ca_V_1.4 knockout mice have presynaptic and postsynaptic morphological abnormalities in the retina that had previously been described. This manuscript reports the use of a knock in mouse model to disambiguate the effects of the Ca_v_1.4 Ca^2+^ channel from those of calcium itself on synapse structure in the OPL. To do so, the authors make use of a KI mouse where the WT Ca_v_1.4 channel is replaced by a channel with a point mutation that renders it unable to conduct calcium. Interestingly, the authors find a small change in the size of and relatively moderate disorganization of presynaptic structures in the KI mouse model, more significant disruptions were observed in postsynaptic cells. The authors conclude that the Ca^2+^ channel protein is needed for proper formation and maintenance of the pre-synaptic structure, but calcium entry through channels are necessary for proper recruitment of post-synaptic partners. Overall, the results are novel and the conclusion is well supported by the data.Specific comments:1) The experiments in this paper focused on the synapses in rods. In WT, rod ribbons are easily distinguishable from cone ribbons based on size and shape. In the KI model, rod ribbons become smaller and one cannot assume that they will maintain their normal shape. Indeed, the authors describe multiple ribbons (some misshapen) in some rods in EM. Is the assumption that all spots are rod ribbons in the KO and KI, but only the crescent shaped ones are rod ribbons in the WT? If some are mis-assigned, how might this affect the quantification of the effects on ribbon size. If they can't be convincingly distinguished, then wouldn't it be fairer comparison to analyze all ribbons in the OPL regardless of genotype?

We regret that we did not include sufficient details regarding our quantitative analyses of ribbons in the Materials and methods section. A key detail relevant to the reviewer’s concern is that we restricted our analyses to rod ribbons using double-labeling with cone-arrestin antibodies to generate 3-D binary masks in order to eliminate the confounding influence of the shorter ribbons that characterize cone terminals. We then used a Matlab-based routine (ObjectFinder) to determine the lengths along the major axis all Ctbp2-labeled structures in the portion of the OPL. Because this was done in confocal z-stacks with more than 30 optical sections (voxel size of 0.09 𝜇𝑚 × 0.09 𝜇𝑚 × 0.2 𝜇𝑚 (𝑋 × 𝑌 × 𝑍)), we were able to capture the full-dimensions of each Ctbp2-labeled structure. Thus, we are confident that we are not mistaking cone ribbons for rod ribbons in the KO or KI, and that our approach allowed us to quantitate differences in the distribution of ribbons and spheres specifically in rod terminals of the 3 genotypes. We have added more details regarding ribbon analysis to the Materials and methods section and Results and Discussion.

2) If there is a reliable way to distinguish rod ribbons from cone ribbons, there seems to be a missed opportunity here to look at effects on cone ribbons and their post-synaptic partners, at least at the level of IHC.

We agree that this is an important future direction, but a bit more complicated to address given the diversity of cone bipolar cells and types of synaptic contacts that cones make with them. To fit with the focus of this short report format, we opted to limit our analyses to the rod phenotype of the KI.

Reviewer #2:This manuscript by Maddox et al. presents a study of Ca_v_1.4 in retinal synapse development. The central issue addressed is differentiating the signaling / organizational role(s) of the channel protein from those of Ca^2+^ influx through the channel during retinal rod photoreceptor synapse development. The authors generated a non-conducting channel (G369i KI) that appears to be trafficked normally to synapses and assessed differences between G369i mutants, Ca_v_1.4 KO, and wild-type animals. They found substantial differences between G369i mutants and Ca_v_1.4 KO retinas, with the G369i mutant more closely resembling the wild-type. The authors suggest, then, that the channel protein serves an organizing role during synapse development.One critical control experiment has not been performed, and this must be included in any revision of the manuscript: although the authors show that rods themselves lack Ca^2+^ currents (Figure 1D), the authors do not provide evidence that the G369i KI rods respond to light and that postsynaptic rod bipolar cells do not (indicating a true deficit of synaptic transmission at the rod to rod bipolar cell synapse). As illustrated in Figure 3, the postsynaptic rod bipolar cells do appear to have developed mGluR6-mediated signaling cascades, but the authors' interpretation of the result, that the postsynaptic signaling can develop in the complete absence of presynaptic glutamate release, is not supported by the data presented.

We agree this is an important control experiment. Thus, we have added ERG analyses in Figure 1GI, which show that while KI rods exhibit normal responses to light (*i.e*., normal a-waves), they show no evidence of b-waves that reflect transmission from rods to rod bipolar cells (RBCs).

As well, the efficiency of Ca^2+^-exocytosis coupling at the rod presynapse is incredibly high, and it is worth noting that Ca^2+^ from sources other than voltage-gated Ca^2+^ channels could give rise to enough glutamate release to modulate synapse development.

This is an excellent point. We added this text to the Discussion:

“Like other presynaptic Ca_v_ channels, Ca_v_1.4 is expected to interact directly or indirectly with a diverse array of proteins (Dolphin and Lee, 2020), some of which are required for ribbon formation in photoreceptors (tom Dieck et al., 2005; Kiyonaka et al., 2012). Thus, the Ca_v_1.4 complex could act as a central scaffold for protein interactions that drive the assembly of the ribbon and associated components of the active zone. Alternatively, but not mutually exclusively, voltagedependent conformational changes in the Ca_v_1.4 protein, which are intact in G369i mutant channels (Figure 1C), could trigger signaling pathways that promote synapse maturation. For example, Ca_v_1.4 could conformationally couple to intracellular Ca^2+^ release channels, as Ca_v_1.1 does in skeletal muscle (Adams *et al.*, 1990), which could represent a Ca^2+^ source needed for synapse development.”

With regard to the pre- and postsynaptic organization of the synapse, as illustrated in Figure 4: it seems a bit counterintuitive to me that the absence of an invaginating synapse would result in a closer apposition of ribeye and mGluR6 proteins, given that mGluR6 normally is located at the dendritic tips inserted in the very center of the invagination. The authors should clarify their explanation of this observation.

We have modified the text accordingly:

“A particularly prominent anomaly of the reconstructed spherules was an alteration in postsynaptic architecture. Normally, within the invagination of mouse rod spherules are two HC neurites and an intervening RBC neurite (Figure 4E). Compared to HCs, the RBC neurite tip and the resident mGluR6 receptors are maintained relatively distant—hundreds of nanometers—from the release site (Rao-Mirotznik *et al.*, 1995). This complex organization is thought to be critical for controlling glutamate release volume and spillover dynamics, allowing for differential activation of distinct glutamate receptor populations (Petralia et al., 2018). For spherules in the OPL as well as in the ONL of G369i KI retina, branching neurites of HCs and RBCs formed non-invaginating triadic contacts close to the anchored presynaptic ribbon (Figure 4F,G), rather than invading rod spherules as in WT retina (Figure 4E). Nevertheless, immunolabeling for mGluR6 remained clustered near ribbons in G369i KI retina (Figure 4C), suggesting that pre- and post-synaptic apposition is maintained even in the absence of the normal spacing between RBC tips and the release sites. In agreement with these findings, mGluR6 labeling was significantly closer to that for ribbons in G369i KI than in WT retina as revealed by nearest-neighbor analyses (Figure 4—figure supplement 2). Thus, while dispensable for postsynaptic partner selection at rod synapses, Ca_v_1.4 Ca^2+^ signals are required for the invaginating arrangement of the corresponding neurites within the rod spherule and their proximity to release sites.”

As well, it is worth commenting that the authors addressed the integrity of the presynaptic active zone by visualizing ribeye and bassoon proteins using fluorescence immunohistochemistry, but neither protein-to my knowledge-interacts directly with Cavs. Is there a reason that the authors did not examine Rim / RimBP or CAST/ELKS expression as an assay of active zone organization? These proteins interact more directly with the Ca^2+^ channels.

The goal of the experiments shown in Figure 2F-G was to show that some major constituents of the presynaptic active zone are intact in G369i KI terminals, rather than to assess the localization of proteins known to interact with Ca_v_1.4. However, we have added labeling for RIM2 to Figure 2I.

Presumably, the G369i KI mutation should not affect the voltage-dependent gating of the channel. Has gating in the mutant been characterized? Is the voltage-sensing role of the channel important in development? These are interesting-to me, at least-questions that might be addressed experimentally or, at a minimum, in a revised discussion.

The biophysical consequences of the G369i mutation have been characterized in the context of Ca_v_1.3 (Baig et al., 2011). Similar detailed studies of this mutation in Ca_v_1.4 are complicated by the notoriously low current amplitudes produced upon transfection of this channel, relative to other Ca_v_ channels, in HEK293T cells. However, we have added experiments showing that gating currents, which represent voltage-sensor movements, are relatively normal in G369i mutant channels (Figure 1C). We have also discussed the potential role of voltage-sensing in the Discussion (see point 2 response, above).

Reviewer #3:This well-written, carefully done study shows that structural features, not calcium influx, of Ca_V_1.4 L-type calcium channels are sufficient for proper organization of many key molecules at rod ribbon synapses. However, formation of the synaptic invagination and proper arrangement of post-synaptic partners requires calcium influx. To perform this study, the authors created knockin mice that express a non-conducting form of Ca_V_1.4. The study was technically proficient and clearly described, with helpful illustrations accompanying each figure. Further work will be needed to understand which particular structural elements are important and how calcium influx regulates formation of the invaginating synapse. Testing the ability of rods to respond to light and transmit those signals to rod bipolar cells is important for interpreting their results. One relatively straightforward way to test this would be to record the A- and B-waves of the electroretinogram (ERG) in their mice.

Done.